# Glycated haemoglobin and fasting plasma glucose tests in the screening of outpatients for diabetes and abnormal glucose regulation in Uganda: A diagnostic accuracy study

**Francis Xavier Kasujja** [1,2]*, **Roy William Mayega**[1], **Meena Daivadanam**[3,4,5], **Elizabeth Ekirapa Kiracho**[6], **Ronald Kusolo**[1], **Fred Nuwaha**[7]

**1** Department of Epidemiology and Biostatistics, School of Public Health, College of Health Sciences, Makerere University, Kampala, Uganda, **2** Chronic Diseases and Cancer Theme, MRC/UVRI and LSHTM Uganda Research Unit, Entebbe, Uganda, **3** Department of Food Studies, Nutrition and Dietetics, Uppsala University, Uppsala, Sweden, **4** Department of Women's and Children's Health, International Maternal and Child Health, Uppsala University, Uppsala, Sweden, **5** Department of Global Public Health, Karolinska Institutet, Solna, Sweden, **6** Department of Health Policy, Planning, and Management, School of Public Health, College of Health Sciences, Makerere University, Kampala, Uganda, **7** Department of Disease Control and Environmental Health, School of Public Health, College of Health Sciences, Makerere University, Kampala, Uganda

* fxkasujja@musph.ac.ug

## Abstract

### Background and objectives

To understand the utility of glycated haemoglobin ($HBA_{1C}$) in screening for diabetes and Abnormal Glucose Regulation (AGR) in primary care, we compared its performance to that of the fasting plasma glucose (FPG) test.

### Methods

This was a prospective diagnostic accuracy study conducted in eastern Uganda. Patients eligible for inclusion were consecutive adults, 30–75 years, receiving care at the outpatient department of a general hospital in eastern Uganda. We determined the sensitivity, specificity and optimum cut-off points for $HBA_{1C}$ and FPG tests using the oral glucose tolerance test (OGTT) as a clinical reference standard.

### Results

A total of 1659 participants underwent FPG testing of whom 310 were also $HBA_{1C}$ and OGTT tested. A total of 113 tested positive for diabetes and 168 for AGR on the OGTT. At recommended cut-off points for diabetes, the $HBA_{1C}$ and FPG tests had comparable sensitivity [69.8% (95% CI 46.3–86.1) versus 62.6% (95% CI 41.5–79.8), respectively] and specificity [98.6% (95% CI 95.4–99.6) versus 99.4% (95% CI 98.9–99.7), respectively]. Similarly, the sensitivity of $HBA_{1C}$ and the FPG tests for Abnormal Glucose Regulation (AGR) at ADA cut-offs were comparable [58.9% (95% CI 46.7–70.2) vs 47.7% (95% CI 37.3–58.4), respectively]; however, the $HBA_{1C}$ test had lower specificity [70.7% (95% CI 65.1–75.8)]

**Data Availability Statement:** All stata data files are available from the Figshare database (10.6084/m9. figshare.20152373).

**Funding:** This study is funded by the Swedish International Development Cooperation Agency (Sida) capacity-building grant to Makerere University 2015-2020 Project HS343 (https:// maksweden.mak.ac.ug/). We also acknowledge the support, both financial and in kind, received from the MRC/UVRI and LSHTM Uganda Research Unit to facilitate the publication of this work. The funders had no role in study design, data collection and analysis, decision to publish, or preparation of the manuscript.

**Competing interests:** The authors have declared that no competing interests exist.

than the FPG test [93.5% (95% CI 88.6–96.4)]. At the optimum cut-offs points for diabetes [45.0 mmol/mol (6.3%) for $HBA_{1C}$ and 6.4 mmol/L (115.2 mg/dl) for FPG], $HBA_{1C}$ and FPG sensitivity [71.2% (95% CI 46.9–87.8) versus 72.7% (95% CI 49.5–87.8), respectively] and specificity [95.1% (95% CI91.8 97.2) versus 98.7% (95% CI 98.0 99.2), respectively] were comparable. Similarly, at the optimum cut-off points for AGR [42.0 mmol/mol (6.0%) for the $HBA_{1C}$ and 5.5 mmol/l (99.0 mg/dl) for the FPG test], $HBA_{1C}$ and FPG sensitivity [42.3% (95% CI 31.8–53.6) and 53.2 (95% CI 43.1–63.1), respectively] and specificity [89.1% (95% CI 84.1 92.7) and 92.7% (95% CI 91.0 94.1), respectively] were comparable.

## Discussion

$HBA_{1C}$ is a viable alternative diabetes screening and confirmatory test to the FPG test; however, the utility of both tests in screening for prediabetes in this outpatient population is limited.

## Introduction

Type 2 diabetes (T2D) is one of the leading public health challenges today. About three out of four (79%) of all people with diabetes live in low- and middle-income countries [1]. Sub-Saharan Africa (SSA) is the region with the highest proportion of undiagnosed diabetes globally; of the 19.4 million people living with diabetes in SSA, 11.6 million (59.7%) are unaware of their condition [2]. Diabetes has been associated with different comorbidities including severe bacterial infections, such as tuberculosis, and viral infections including severe acute respiratory syndrome coronavirus 2 (SARS-CoV-2) [3]. As the world faces an unprecedented SARS-CoV-2 pandemic, improving diabetes diagnosis could not be overemphasized [4].

Individuals with undiagnosed diabetes develop uncontrolled hyperglycemia which puts them at high risk of developing micro- and macrovascular T2D diabetes complications. The diagnostic gaps in related to T2D seen in SSA are due to a low awareness of T2D risk factors and poor access to diabetes testing facilities among the general population, limited knowledge of screening and diagnosis among health professionals, and limited access to low cost screening tests [5, 6].

The two tests commonly used for T2D testing are the Fasting Plasma Glucose (FPG) and the Glycated Haemoglobin test (HBA1C). The HBA1C test has a major practical advantage over the FPG test in that it does not require overnight fasting. Because of this, test results can be availed to the patient at their index health facility visit. The HBA1C test also measures the average glycaemia over the preceding 2–3 months [7], unlike the FPG test which measures glycemia at a point in time. It also has less intra-individual variability [8], greater analytical stability, and is a better predictor of microvascular disease than FPG [9].

However, HBA1C levels vary by race [10, 11], but no such racial variation in FPG levels has been reported. In addition, red blood cell lifespan determines the duration of exposure to glucose. Therefore, conditions that increase red cell turnover–such as haemoglobinopathies, malaria, and haemolytic anaemias–are associated with spuriously low $HBA_{1C}$ results [12]. Other conditions, such as iron deficiency anaemia (IDA), on the contrary, are associated with elevated $HBA_{1C}$ levels [13, 14].

Moreover, the WHO-recommended $HBA_{1C}$ cut-offs have been derived from predominantly non-black populations. It is not clear whether these figures could be extrapolated to

geographical areas with higher prevalence of heamoglobinapathies, malaria and anaemia such as SSA [15], like Uganda. Sickle cell disease, malaria and other parasitic infections are endemic, with a third of the women aged 15–49 years and 16% of the men anaemic [16]. Validating screening tests for use in sub-Saharan Africa is crucial for tackling the rising type 2 diabetes epidemic in the region. In this study, we compared the diagnostic performance of point-of-care (POC) $HBA_{1C}$ and FPG tests when used to screen for diabetes and prediabetes in an outpatient population.

## Methods

### Setting

The study was conducted at the outpatient department of Iganga general hospital in eastern Uganda, 120 kilometres from Kampala, the capital city of Uganda. The hospital serves communities in Iganga and surrounding districts, whose main economic activities are subsistence farming, fishing, and petty trading. In addition to inpatient services, the hospital operates a busy outpatient department (OPD) for patients referred from lower-level health facilities; however, most patients self-refer to the hospital. The outpatient department provides general preventive, promotional, curative, and maternity services. Patients suspected of having diabetes in the OPD are referred to the diabetes clinic, which is conducted every Tuesday.

### Study design

A diagnostic accuracy study was conducted. Eligible participants were patients aged 30–75 years accessing care at the OPD of Iganga hospital. Individuals who were known to have diabetes and on anti-hyperglycaemic drugs, anti-psychotic drugs, systemic steroids, those known to have sickle cell disease or with clinical features suggestive of sickle cell disease, and those with a history of having undergone blood transfusion within the previous 3 months were excluded.

A sample of 110 participants with diabetes was required to demonstrate noninferiority of the $HBA_{1C}$ test compared to the FPG for a minimum relative sensitivity of 0.75 for the two tests with a power of 80% and a significance level of 0.05, assuming a sensitivity of 78% and 81% for the $HBA_{1C}$ and the FPG test [17], respectively.

We used a two-stage sampling strategy. The first stage involved consecutive sampling and FPG testing of all eligible study participants. At the second stage, test result-based sampling [18] was conducted such that only a sub-sample of the study participants underwent both the $HBA_{1C}$ test and verification with a two-hour oral glucose tolerance test (OGTT) clinical reference standard. This sampling was done among individuals with more than 6.0 mmol/l and those with less than or equal to 6.0 mmol/l in a 1:1 ratio. The two tests were conducted in every participant who scored greater than 6.0 mmol/l on the first stage FPG test. In addition, the next study participant who scored 6.0 mmol/l or less on the first stage FPG test was also $HBA_{1C}$ and OGTT tested to obtain the 1:1 ratio (*Fig 1*). The approach ensured that the sample of individuals undergoing all three tests had higher proportion of individuals with hyperglycaemia and diabetes. This way, the sample size could be attained at a minimal cost. The two-hour OGTT clinical reference was chosen because it is is a more sensitive than either the FPG or $HBA_{1C}$ tests and better at mimicing postprandial hyperglycemia, the earliest form of dysglycemia [19, 20].

### Data collection procedures

The data collection forms were programmed using Open Data Kit (ODK) platform. Data was collected by research assistants (nurses and laboratory technologists) experienced in diabetes

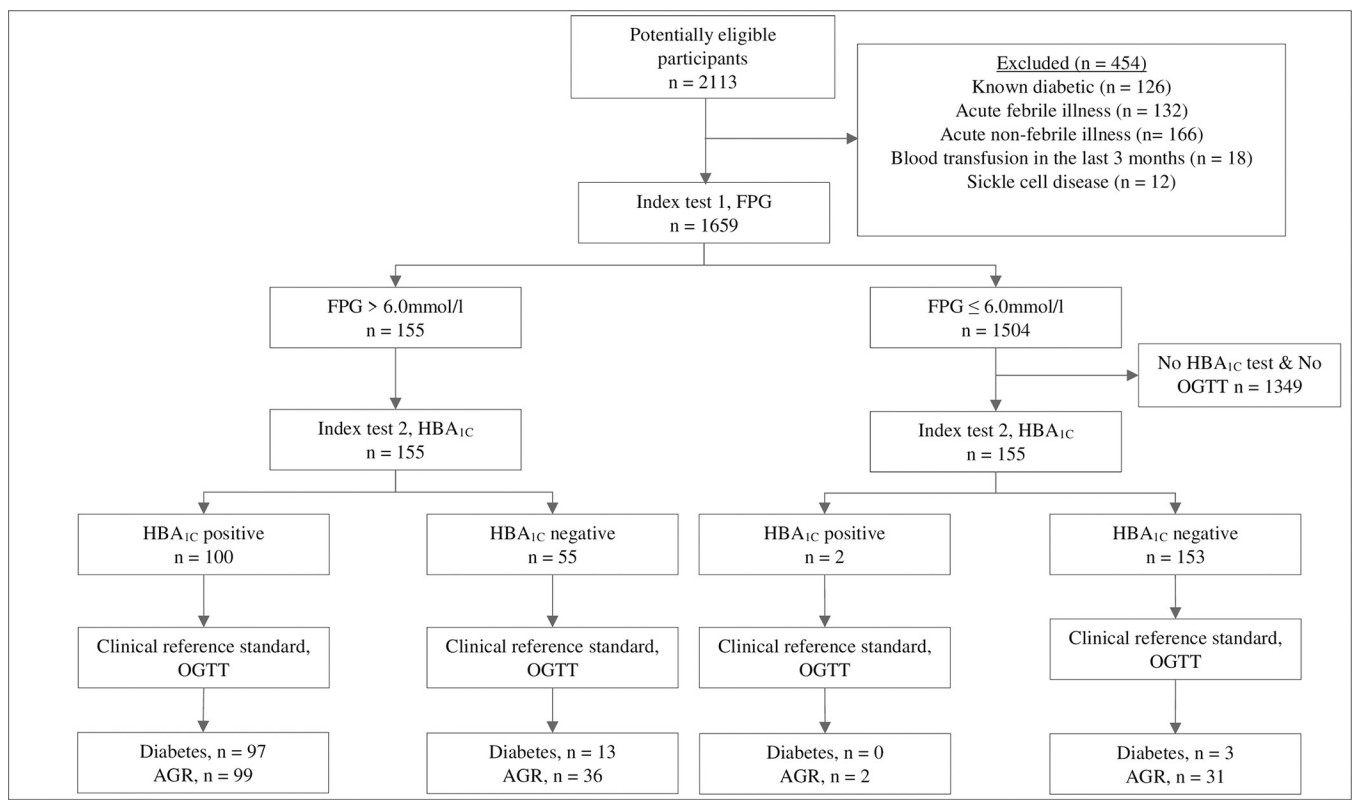

**Fig 1. Flow of participants through the study.**

screening who used android-based tablets. All data entered on the tablet was transmitted to a secure server via internet. The study participants were recruited from the waiting area of the outpatient department. Eligible study participants were requested to return to the hospital the next morning having had their last meal at dinnertime.

A structured questionnaire was used to collect sociodemographic data such as on age, sex, and education level. Anthropometric measurements were made following standard procedures. Height was measured to the nearest centimetre using seca® 213 stadiometers (seca gmbh, Germany), with the participant standing upright. Weight was measured to the nearest 0.1 kg using a seca® 813 (seca gmbh, Germany) weighing scale, with the participant wearing light clothing without shoes. Waist circumference was measured in centimetres using a tape measure strapped along a horizontal plane at the midpoint between the bottom of the patient's last rib and the top of the iliac crest. Body mass index (BMI) was calculated using the formula, weight in kg/(Height in m)$^2$. The blood pressure was determined from the mean of three readings measured 10 minutes apart using an Omron M2 blood pressure monitor (model HEM-7121-E, Omron Corporation) with the patient seated.

## Laboratory investigations

In all cases, the FPG and HBA1C were conducted by the same research assistants before the OGTT. The FPG test was performed on each consenting adult from a drop of capillary blood drawn from a finger prick using a spring-loaded lancet. The blood was analysed using the handheld Accu-Chek® Active blood glucose meter (Roche Diagnostics GmbH, Mannheim, Germany), which is readily available on the Ugandan market. This glucometer meets ISO

15197:2013 specifications, according to the manufacturer's product insert. This means that at least 95% of the results of this glucometer are within ±15 mg/dl at glucose concentrations <100 mg/dl and within ±15% at ≥100 mg/dl compared to a traceable laboratory method [21]. For the OGTT, participants drank 75 g of anhydrous glucose dissolved in 250 mls of water after which they were asked to sit on a bench within the testing area. After 2 hours, a capillary blood sample was taken by finger prick and tested with the glucometer. Similarly, the $HBA_{1C}$ test was done on capillary blood from a finger prick using a Cobas b 101 system (Roche Diagnostics GmbH, Mannheim, Germany), a benchtop device accredited by the National Glycohemoglobin Standardization Program (NGSP) and standardized to the Diabetes Control and Complications Trial (DCCT) assay. The coefficient of variation of the Cobas b 101 instrument is less than 5% [22], as recommended by the International Federation of Clinical Chemistry and Laboratory Medicine for $HBA_{1C}$ POC tests [23, 24].

## Definition of diabetes and abnormal glucose regulation

The Standards for Reporting of Diagnostic Accuracy Studies (STARD) 2015 checklist [25] was followed with FPG and $HBA_{1C}$ as index tests and OGTT as the reference standard. Diabetes was defined based on OGTT ≥11.1 mmol/L, FPG ≥7.0 mmol/L, and $HBA_{1C}$ ≥48 mmol/mol, per World Health Organization (WHO) [26] and the American Diabetes Association (ADA) criteria (ADA 2021). The term 'Abnormal Glucose Regulation' (AGR) was used to define a composite of prediabetes and diabetes based on the following criteria: OGTT ≥ 7.8 mmol/L (WHO and ADA criteria), FPG ≥6.1 mmol/L (WHO criteria) and $HBA_{1C}$ ≥ 39 mmol/mol (ADA criteria).

## Statistical analysis

Continuous variables were presented as mean and categorical variables were presented as frequencies and proportions. Subjects were divided into 3 groups based on whether they had normal glycaemia, AGR or diabetes, according to OGTT criteria.

The diagnostic accuracy of $HBA_{1C}$ and FPG to diagnose diabetes and abnormal glucose regulation when OGTT was used as the reference standard was assessed on seven dimensions: sensitivity, specificity, positive likelihood ratio and negative likelihood ratio, positive predictive value (PPV), negative predictive value (NPV) and interval likelihood ratios. These were computed after verification bias-correction of test results using inverse probability weighting (Pepe 2003) (See S1–S4 Appendices). We explored how the sensitivity and specificity of the FPG and $HBA_{1C}$ would vary in a simultaneous testing using a scenario tree.

Receiver operating characteristic (ROC) curves were plotted using Stata version 14 and the area under curves (AUC) for both the $HBA_{1C}$ and FPG tests determined using the R statistical package. P-values ≤ 0.05 were considered statistically significant. We considered the optimum cut-offs for the two tests as the points on their ROC curves where sensitivity and specificity were highest.

## Ethical considerations

The study was approved by Makerere University School of Public Health Higher Degrees, Research and Ethics Committee and the Uganda National Council of Science and Technology (reference number: HS 2611). Administrative clearance was sought and received from Iganga Hospital. All participants provided written informed consent. To ensure confidentiality, anonymous identifiers were used during data collection, all tablets used for data collection were password protected, and data was encrypted during transmission to prevent unauthorized

access. All study participants diagnosed with diabetes during the study were referred to the Iganga Hospital diabetes clinic for further management.

## Results

The characteristics of the participants who underwent testing with both index tests and the clinical reference standard are summarized in *Table 1*. The mean age was 49.0 years (95% CI 47.8–50.2). The mean BMI was 26.8 kg/m$^2$ (95% CI 26.2–27.4). Women had a higher mean BMI [27.4 kg/m$^2$ (95% CI 26.7–28.1)] compared to men [24.8 kg/m$^2$ (95% CI 23.5–25.9)]. Similarly, a higher proportion of women [27.4% (95% CI 26.7–28.1] reported eating vegetables and fruits daily compared to men [24.8% (95% CI 23.5–25.9)] (*Table 1*).

The Standards for Reporting of Diagnostic Accuracy Studies flow diagram is shown in *Fig 1*, 2113 subjects were screened for study eligibility between 6 January 2020 and 26 February 2021. Of these, 1659 participants consented to participate in the study and underwent an initial FPG test. Those whose FPG results were greater than 6.0 mmol/l (155/1659) underwent HBA$_{1C}$ and OGTT testing and 110 tested positive for diabetes on the OGTT (*Fig 1*). Similarly, 155 of those whose FPG results were less or equal to 6.0 mmol/l were also HBA$_{1C}$ and OGTT tested and 3 tested positive for diabetes on the OGTT.

### Sensitivity and sensitivity as screening tests

The sensitivity of the FPG and HBA$_{1C}$ tests when screening for diabetes based on WHO and ADA criteria [69.8% (95% CI 46.3–86.1)] versus 62.6% (95% CI 41.5–79.8), respectively] were comparable and moderate (*Table 2*). Similarly, the specificity of the FPG and HBA$_{1C}$ tests were not only comparable [99.4% (95% CI 98.9–99.7) versus 98.6% (95% CI 95.4–99.6), respectively] but high. The positive predictive value (PPV) of the FPG test (0.91) was substantially higher than that of HBA$_{1C}$ test (0.82). The sensitivity of the HBA$_{1C}$ test when screening for AGR according to ADA criteria [58.9% (95% CI 46.7%-70.2%)] was only moderate, and was comparable to that of the FPG test [47.7% (95% CI 37.3%-58.4%)]. However, the specificity of the FPG test in detecting AGR [93.5% (95% CI 88.6%-96.4%)] was substantially greater than that of the HBA$_{1C}$ test [70.7% (95% CI 65.1%-75.8%)] (see Table 2). The sensitivity and

**Table 1. Descriptive statistics.**

| Characteristics | Total (N = 310) | Men (N = 69) | Women (N = 241) | p-value |
|---|---|---|---|---|
| Age (years) | 49.0 (47.8, 50.2) | 51.0 (48.1, 53.8) | 48.4 (47.1, 49.7) | 0.08 |
| BMI (kg/m$^2$)* | 26.8 (26.2, 27.4) | 24.8 (23.5, 25.9) | 27.4 (26.7, 28.1) | <0.00 |
| Waist circumference (cm) | 89.9 (88.4, 91.5) | 87.1 (83.9, 90.4) | 90.73 (89.0, 92.5) | 0.06 |
| Systolic blood pressure (mmHg) | 133.6 (131.1, 136.1) | 132.6 (127.6, 137.7) | 133.9 (130.9, 136.8) | 0.69 |
| Diastolic blood pressure (mmHg) | 88.0 (86.5, 89.6) | 87.1 (83.7, 90.5) | 88.3 (86.5, 90.1) | 0.51 |
| FPG (mmol/l) | 7.4 (6.9, 7.9) | 8.3 (7.0, 9.6) | 7.2 (6.6, 7.7) | 0.10 |
| 2-hour PG (mmol/mol) | 11.4 (10.5, 12.2) | 12.0 (10.1, 13.9) | 11.2 (10.2, 12.2) | 0.46 |
| HBA$_{1C}$ (mmol/mol) | 49.9 (47.2, 52.6) | 53.1 (47.0, 59.2) | 49.0 (46.0, 52.0) | 0.21 |
| Family history of diabetes (%) | 21.0 (16.6, 25.9) | 15.9 (7.3, 24.6) | 22.4 (17.1, 27.7) | 0.25 |
| Engages in least 30 minutes of physical activity per day (%) | 94.2 (91.0, 96.5) | 92.8 (86.6, 98.9) | 94.6 (91.8, 97.5) | 0.56 |
| Eats vegetables and fruits (%)* | 14.8 (11.1, 19.3) | 7.3 (1.1, 13.4) | 17.0 (12.3, 21.8) | 0.04 |

Data are mean (95% CI) or percentage (95% CI). Comparisons of characteristics between men and women were done using t-tests for continuous variables and proportion tests for categorical variables.

*P-value < 0.05

**Table 2. Sensitivity, specificity, PPV and NPV of the FPG and HBA$_{1C}$ tests in the diagnosis of diabetes and AGR at the OGTT cut-offs recommended by ADA and WHO.**

| | Test | Sensitivity, %<br>(95% CI) | | Specificity, %<br>(95% CI) | | PPV | | NPV | |
|---|---|---|---|---|---|---|---|---|---|
| Diabetes | FPG | 62.6 (41.5–79.8) | | 99.4 (98.9–99.7) | | 0.91 | | 0.97 | |
| | HBA$_{1C}$ | 69.8 (46.3, 86.1) | | 98.6 (95.4–99.6) | | 0.82 | | 0.97 | |
| AGR | | Based on ADA criteria | Based on WHO criteria | Based on ADA criteria | Based on WHO criteria | Based on ADA criteria | Based on WHO criteria | Based on ADA criteria | Based on WHO criteria |
| | FPG | 47.7 (37.3–58.4) | 29.0 (22.5–36.6) | 93.5 (88.6–96.4) | 98.3 (97.4–98.9) | 0.74 | 0.87 | 0.82 | 0.78 |
| | HBA$_{1C}$ | 58.9 (46.7–70.2) | NA | 70.7 (65.1–75.8) | NA | 0.45 | NA | 0.82 | NA |

specificity of the FPG test in detecting AGR according to the WHO criteria were 29.0% (95% CI 22.5%-36.6%) and 98.3% (95% CI 97.4%-98.9%), respectively.

## Optimal cut-off values

Fig 2 shows the ROC curves for FPG and HBA1C in screening for Diabetes and AGR. The FPG and HBA1C tests had comparable discriminatory capacity for both diabetes (AUC = 0.85 versus 0.86, respectively) and AGR (AUC = 0.74 versus 0.68, respectively).

The optimum diabetes cut-offs for the HBA$_{1C}$ and FPG for diabetes were 45 mmol/mol (6.3%) and 6.40 mmol/L (115 mg/dL) (Table 3); the sensitivities and specificities for the two tests at these cut-off points (71.2% and 98.7%, respectively, for the HBA$_{1C}$ test and 72.7% and 98.7%, respectively, for the FPG) were comparable. On the other hand, the optimum AGR cut-offs for the HBA$_{1C}$ and the FPG test were 42.0 mmol/mol (6.0%) and 5.50 mmol/L (99 mg/dL), respectively. The sensitivity and specificity for the HBA$_{1C}$ test (42.3% and 89.1%, respectively) and FPG (53.2% and 92.7%, respectively) at these points were similarly comparable.

## Discussion

The question of whether the HBA$_{1C}$ test performs adequately compared to the FPG test when used to screen for diabetes and prediabetes in SSA is fundamental to its wider uptake in the

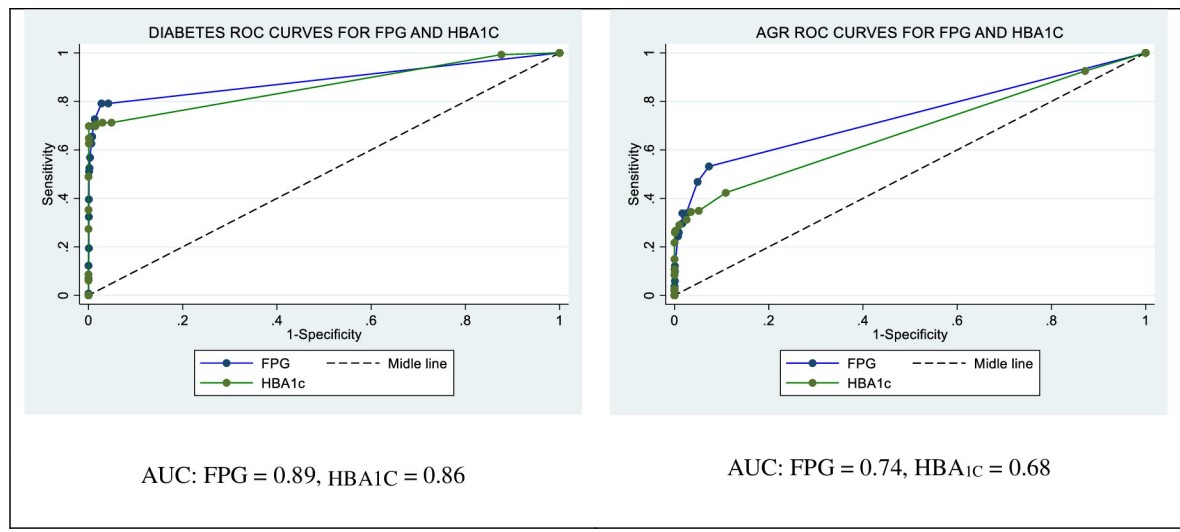

AUC: FPG = 0.89, HBA1C = 0.86          AUC: FPG = 0.74, HBA1C = 0.68

**Fig 2. ROC curves for FPG and HBA$_{1C}$ in the diagnosis of diabetes and AGR.**

**Table 3. Sensitivity and specificity at standard and optimum cut-off points.**

| Hyperglycaemic state | Test | Cut-off points[‡] | | Sensitivity (95% CI)[#] | | Specificity (95% CI)[#] | |
|---|---|---|---|---|---|---|---|
| | | Standard | Optimum | Standard | Optimum | Standard | Optimum |
| Diabetes | FPG | ≥7.0 (126) | 6.4 (115.2) | 62.6 (41.5–79.8) | 72.7 (49.5–87.8) | 99.4 (98.9–99.7) | 98.7 (98.0 99.2) |
| | HBA$_{1C}$ | ≥48 (6.5) | 45.0 (6.3) | 69.8 (46.3, 86.1) | 71.2 (46.5–87.6) | 98.6 (95.4–99.6) | 95.1 (91.8–97.2) |
| AGR | FPG (WHO) | ≥ 6.1 (110) | 5.5 (99.0) | 29.0 (22.5–36.6) | 53.2 (43.1–63.1) | 98.3 (97.4–98.9) | 92.7 (91.0–94.1) |
| | FPG (ADA) | ≥ 5.6 (100) | | 47.7 (37.3–58.4) | | 93.5 (88.6–96.4) | |
| | HBA$_{1C}$ (ADA) | ≥ 39 (5.7) | 42.0 (6.0) | 58.9 (46.7–70.2) | 42.3 (31.8–53.6) | 70.7 (65.1–75.8) | 89.1 (84.1–92.7) |

[‡]Units are mmol/L (mg/dl) for FPG and mmol/mol (%) for HBA$_{1C}$

[#]Data are %

region [27]. In this validity study, which was conducted at a general hospital in eastern Uganda, generally the performance of the HBA$_{1C}$ test in screening for diabetes and AGR was comparable to that of the FPG test. However, the specificity of the HBA$_{1C}$ test in screening for AGR at ADA recommended cut-offs was lower than that of the FPG. This is one of the few FPG and HBA$_{1C}$ validity studies conducted among a SSA outpatient population that used an OGTT clinical reference standard.

The sensitivity of the HBA$_{1C}$ when screening for diabetes was moderate at standard cut-off points. Hird et al [28] found a similar HBA$_{1C}$ diabetes sensitivity of 70.3% among a black South African general population in the Durban Diabetes Study. However, an earlier study conducted among a mixed ancestry South African population, the Bellville-South cohort [29], found an even lower HBA$_{1C}$ sensitivity estimate for diabetes (45.9%). The disparity in HBA$_{1C}$ sensitivity evident in these studies could be explained by differences in patient spectrum affecting both test accuracy and prevalence [30]. The mixed ancestry population had a much higher diabetes prevalence (46.9%) [29] compared to the black South African population (12.9%) [28] and our study population (8.4%). Besides, our study was conducted among outpatients, a population likely to have major clinical differences compared to the general populations that were the focus of the other two studies. The sensitivity of the HBA$_{1C}$ test was higher than that of the FPG test in this study which points to the potential for the HBA$_{1C}$ test to correct detect a higher proportion of individuals with diabetes than the FPG test for an outpatient population in this setting at standard cut-off points. Used together in a simultaneous testing scenario, the two tests could have a combined sensitivity as high as 88.7% at standard cut-off points. However, testing individuals with both tests in this way would require considerable resources and may only be possible in some cases.

The HBA$_{1C}$ test had a high specificity at standard cut-off points (98.4%), which was comparable to that of the FPG test (99.4%). This finding is consistent with that of Zemlin et al [29] and Hird et al [28] who found HBA$_{1C}$ specificities of 96% and 98.7%, respectively. The high specificity of both the HBA$_{1C}$ and FPG tests highlights the minimal number of individuals with false positive diabetes diagnoses detected by either test. This finding supports the utility of the HBA$_{1C}$ test, alongside the FPG test, as confirmatory tests.

At the optimum cut-off point, 45.0 mmol/mol (6.3%), which was lower than the WHO and ADA recommended threshold, the HBA$_{1C}$ test had a specificity of 95.1%. This threshold was higher than that reported by Zemlin et al [29] and Hird et al [28] (77% and 92%, respectively) from South Africa who found lower cut-off points [43 mmol/mol (6.1%) and (≥42 mmol/mol [6.0%]), respectively]; however, the two studies reported higher sensitivities (77% and 92%, respectively) compared to our study (71.2%). These differences may be attributed to the patient spectrum differences discussed in the previous paragraph for the three studies at standard

diabetes cut-off points. Also notable is that the optimum FPG cut-off point [6.4 mmol/L (115 mg/dl)] was also lower than the WHO and ADA recommended threshold. The sensitivity (72.7%) and specificity (98.7%) of the FPG test at this point were comparable to those of the $HBA_{1C}$ test. The $HBA_{1C}$ and FPG tests could potentially be better at detecting diabetes at their optimal cut-off points than the currently recommended cut-offs.

The FPG test had peak AGR sensitivity (53.2%) and a high specificity (98.7%) at its optimal cut-off point [5.5 mmol/L (99 mg/dl)]. This FPG cut-off point was lower than the threshold recommended by the the WHO [6.1 mmol/L (110 mg/dl)] but close to the ADA cut-point [5.6 mmol/L (100 mg/dl)]. Compared to the WHO recommended cut-off point, such a low FPG cut-off point could lead to overdiagnosis of AGR with the potential to overwhelm the under-resourced public health system [31]. For the $HBA_{1C}$ test, the optimum cut-off point [$HBA_{1C} \geq$ 42 mmol/mol (6.0%)] was higher than the ADA recommended cut-off point [$HBA_{1C} \geq 39$ mmol/mol (5.6%)] and similar to the threshold recommended by the International Expert Committee [32]. Therefore, fewer people would be correctly diagnosed with AGR at the optimum $HBA_{1C}$ cut-off compared to the number that would be at the $HBA_{1C}$ ADA cut-off point. Using the two tests together in a simultaneous testing scenario for AGR at the WHO cut-off for the FPG test and the optimum cut-off for the $HBA_{1C}$ test would lead to only a slightly higher combined sensitivity of 59%. However, the combined sensitivity would be much higher (81%) for a simultaneous testing strategy involving the FPG test at its optimum cut-off point and the $HBA_{1C}$ at the ADA-recommended cut-off point.

The $HBA_{1C}$ test seems like a viable alternative to the FPG test for the screening of type 2 diabetes among the outpatient population, based on the findings of this study. The $HBA_{1C}$ test is particularly suited for screening in the outpatient population as it is minimally affected by acute illness [33] or short-term physical exertion, as may occur when patients walk or ride to the health facility. As it does not require fasting, it would be available to patients throughout the day and night. To improve the yield and positive predictive value of the test, the $HBA_{1C}$ test could be targeted to high-risk individuals. Obesity, physical inactivity and poor dietary diversity are some of the known diabetes risk factors in the general population in eastern Uganda [34]. A growing body of evidence points to differences in the risk profiles of diabetes in SSA compared to populations elsewhere; for example, the early onset of diabetes in SSA and the large burden of disease among individuals with diabetes with normal BMI [34, 35]. Optimal diabetes screening strategies for Uganda and similar settings should be based on a better understanding of the appropriate diabetes risk strata to ensure efficiency. We found the $HBA_{1C}$ to have the potential to correctly detect more individuals with diabetes and prediabetes in this population. Since this test is more costly than the FPG test any screening strategy based on the $HBA_{1C}$ test would call for further studies to compare its cost-effectiveness to that of the FPG test.

$HBA_{1C}$ POC testing is a viable alternative to FPG testing in the screening of diabetes in the outpatient setting. Both tests had high specificity for diabetes and could therefore be used for second-line confirmatory screening of individuals who test positive. The sensitivity for both tests for testing for AGR and therefore prediabetes are low, limiting their utility in the screening of prediabetes.

## Recommendations

Our study has implications on the testing of diabetes in Uganda and similar contexts. Providing $HBA_{1C}$ testing to complement or as an alternative to FPG screening at the health facility level has the potential to improve the detection of diabetes. The FPG test could be provided to patients who arrive at the facility early and are still in a fasted state, reserving the $HBA_{1C}$ test

for those who arrive later in the day. Either test could be used for confirmatory testing for diabetes. However, confirmatory testing for prediabetes should be reserved for the FPG at the WHO cut-off point. Where resources allow, a combination of the $HBA_{1C}$ test and the FPG tests could be used to screen both diabetes and prediabetes. Future studies should evaluate the performance of the $HBA_{1C}$ and FPG in sub-populations with high red cell turnover, and their effectiveness and cost effectiveness in the targeted screening of diabetes and prediabetes among high risk populations.

## Study limitations

Previous research has shown that HIV patients on protease inhibitors and nucleoside reverse transcriptase inhibitors have a different diabetes risk profile from the general population, our study was underpowered to evaluate the performance of the $HBA_{1C}$ in this patient population. Even so, previous research in the Uganda population [36] found that the FPG and $HBA_{1C}$ tests correlation among HIV patients was similar to that among members of the general population (r = 0.69 vs. r = 0.66). Similarly, there was no evidence of a statistically difference between mean FPG and $HBA_{1C}$ levels between the two groups. However, a study conducted in a population of HIV patients on antiretroviral therapy in South Africa [37] found a very low sensitivity of 37% at the WHO recommended cut-of.

We did not evaluate $HBA_{1C}$ performance among individuals with haemoglobin variants, such as sickle cell trait, which could interfere lead to diabetes misclassification [38]. However, the Cobas b101 benchtop analyzer used in our study is stable to sickle cell trait interference [23] and would be expected to perform to the same level in this patient category. We were not able to assess the validity of the $HBA_{1C}$ test for sub-populations with anaemia and other conditions leading to high red blood cell turnover, as our study was not powered for this analysis. Similarly, we did not collect data on the pregnancy status of the female participants of this study which is another limitation of our findings. Finally, the two-hour OGTT reference only measures one of several dysglycaemic aetiologies, limiting its utility as a diabetes and prediabetes clinical reference standard.

## Supporting information

**S1 Checklist.**
(DOCX)

**S2 Checklist. STARD for abstracts: Essential items for reporting diagnostic accuracy studies in journal or conference abstracts.**
(DOCX)

**S1 Appendix. Calculations for the sensitivity and specificity of the FPG and $HBA_{1C}$ tests when used to screen for diabetes.**
(DOCX)

**S2 Appendix. Calculations for the sensitivity and specificity of the FPG and $HBA_{1C}$ tests when used to screen for AGR.**
(DOCX)

**S3 Appendix. Computation of sensitivity and specificity for optimum cut-offs for Diabetes and the corresponding 95% CIs.**
(DOCX)

**S4 Appendix. ROC and interval likelihood ratios for the FPG and HBA$_{1C}$ tests when used to screen for diabetes and AGR.**
(DOCX)

## Author Contributions

**Conceptualization:** Francis Xavier Kasujja, Roy William Mayega, Meena Daivadanam, Elizabeth Ekirapa Kiracho, Fred Nuwaha.

**Data curation:** Francis Xavier Kasujja, Ronald Kusolo.

**Formal analysis:** Francis Xavier Kasujja, Roy William Mayega, Ronald Kusolo.

**Funding acquisition:** Roy William Mayega, Elizabeth Ekirapa Kiracho.

**Investigation:** Francis Xavier Kasujja, Ronald Kusolo.

**Methodology:** Francis Xavier Kasujja, Roy William Mayega, Meena Daivadanam, Elizabeth Ekirapa Kiracho, Fred Nuwaha.

**Project administration:** Francis Xavier Kasujja.

**Software:** Francis Xavier Kasujja, Ronald Kusolo.

**Supervision:** Roy William Mayega, Meena Daivadanam, Fred Nuwaha.

**Validation:** Roy William Mayega, Meena Daivadanam, Elizabeth Ekirapa Kiracho, Fred Nuwaha.

**Visualization:** Roy William Mayega, Meena Daivadanam.

**Writing – original draft:** Francis Xavier Kasujja.

**Writing – review & editing:** Francis Xavier Kasujja, Roy William Mayega, Meena Daivadanam, Elizabeth Ekirapa Kiracho, Ronald Kusolo, Fred Nuwaha.

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
