## [Decision Letter · Decision Letter 0]

25 May 2022

PONE-D-22-06422Glycated Haemoglobin and Fasting Plasma Glucose tests in the screening of outpatients for diabetes and abnormal glucose regulation in Uganda: a diagnostic accuracy studyPLOS ONE

Dear Dr. Francis Xavier Kasujja

Thank you for submitting your manuscript to PLOS ONE. After careful consideration, we feel that it has merit but does not fully meet PLOS ONE’s publication criteria as it currently stands. Therefore, we invite you to submit a revised version of the manuscript that addresses the points raised during the review process.An interesting and relevant work to inform a key research gapThe manuscript will benefit more by addressing the followingLin 205 to 208 authors will need reference the table which describes these results.It is not clear as it is 155 participants are those with above 6mmol/L or "above or less"In the result section authors have described findings of the outcomes of the study before description of demographics and clinical characteristics. Authors will need to consistently follow the STARD guideline and use the STARD checklist to REWRITE their manuscript.In table 2 authors are reporting specificity that is comparable (overlapping CI) however in the result description and summary in the discussion, authors have reported low specificity of HBA1c as low specificity. Authors will need to clarify or rectify.Please submit your revised manuscript by Jul 09 2022 11:59PM. If you will need more time than this to complete your revisions, please reply to this message or contact the journal office at plosone@plos.org. Please include the following items when submitting your revised manuscript:A rebuttal letter that responds to each point raised by the academic editor and reviewer(s). You should upload this letter as a separate file labeled 'Response to Reviewers'.A marked-up copy of your manuscript that highlights changes made to the original version. You should upload this as a separate file labeled 'Revised Manuscript with Track Changes'.An unmarked version of your revised paper without tracked changes. You should upload this as a separate file labeled 'Manuscript'.If applicable, we recommend that you deposit your laboratory protocols in protocols.io to enhance the reproducibility of your results. Protocols.io assigns your protocol its own identifier (DOI) so that it can be cited independently in the future. For instructions see: https://journals.plos.org/plosone/s/submission-guidelines#loc-laboratory-protocols. Additionally, PLOS ONE offers an option for publishing peer-reviewed Lab Protocol articles, which describe protocols hosted on protocols.io. Read more information on sharing protocols at https://plos.org/protocols?utm_medium=editorial-email&utm_source=authorletters&utm_campaign=protocols.

We look forward to receiving your revised manuscript.

Kind regards,

Fredrick Baragi Haraka, MD, PhD

Academic Editor

PLOS ONE

Journal Requirements:

Additional Editor Comments :

An interesting and relevant work to inform a key research gap

The manuscript will benefit more by addressing the following

Lin 205 to 208 authors will need reference the table which describes these results.It is not clear as it is 155 participants are those with above 6mmol/L or "above or less"

In the result section authors have described findings of the outcomes of the study before description of demographics and clinical characteristics. Authors will need to consistently follow the STARD guideline and use the STARD checklist to REWRITE their manuscript.

In table 2 authors are reporting specificity that is comparable (overlapping CI) however in the result description and summary in the discussion, authors have reported low specificity of HBA1c as low specificity. Authors will need to clarify or rectify.

Reviewers' comments:

Reviewer's Responses to Questions

**Comments to the Author**

1. Is the manuscript technically sound, and do the data support the conclusions?

Reviewer #1: Yes

2. Has the statistical analysis been performed appropriately and rigorously? 

Reviewer #1: No

3. Have the authors made all data underlying the findings in their manuscript fully available?

Reviewer #1: Yes

4. Is the manuscript presented in an intelligible fashion and written in standard English?

Reviewer #1: Yes

5. Review Comments to the Author

Reviewer #1: Interesting piece of science given the near absence of data on diagnostic utility of current recommended diabetes screening and diagnostic tools among Africans.

The choice of the topic is also of clinical significance to many physicians practicing in sub-Saharan Africa. Besides, their otherwise large sample size is a good indication of effort made to inform science on this important but otherwise neglected topic. However, after reading through the manuscript, I have several queries:

1. Are investigators in a position to run 'sensitivity analysis' of their findings prior to publishing the manuscript?

Reason: The investigators reported in their manuscript to had applied 2-stage sampling strategies. However, in both cases, the approaches were non-probability sampling in nature. Strangely, the inferences were heavily dependent on probability sampling analysis. For all practical purposes, the so called "consecutive sampling" is a non-probability sampling method. One of the immediate impacts (despite all posteriori statistical manoeuvres done) of such 'design error' include estimates that blow-out disproportionately out of the bounded confidence interval values. That can be evident in their manuscript since optimum level specificity for their HbA1c had an estimate (98.7%) outside the stated 95% C.I. (96.2% - 98.4%) in both abstract (refer line 47 of page 2) and even in their table 3.

Usually this thinking invalidate the entire analysis of their primary analysis.

However, I am optimistic of the findings, based on both 'near-infinitely large sample size', as well as 'inverse probability weighting', that controlled for bias in the primary findings (sensitivity, specificity, PPV & NPV), obtained out of likely 'partial verification bias' prominent by design. It is highly likely, that their large sample size, approximate the 'real population' of interest, and therefore somehow offset all the doubted possibilities of significant biases. A sensitivity analysis is likely warranted in this scenario.

2. The study design was better suited as population-based study instead of hospital-based analysis of present.

2.1 - as stated by investigators, Iganga hospital is a referral health facility and hence the studied sample is unlikely to be representative of the entire population.

In fact, there is a possibility that their findings to be mere 'hospital statistics' data. Time after time, hospital statistics data are known to be 'negatively biased' with respect to the clinical diagnostic utility studies. No wonder, their sample has an age range of 30-75 years. Highly likely those 'older than 75 years cohort' who are the most at risk to be diabetic were selected out from participating in this study. Likewise, the 'younger cohort' (<30 years) who are otherwise, least likely to be impaired glucose tolerant/diabetic were unlikely to be part of this study

2.2. Prevalent clinical conditions in the population that can significantly affect HbA1c levels were not reported to had been controlled in the study by design.

For instance, how many (and proportion by %) of their participants in the study were pregnant women?

and how many (and proportion by %) were HIV-seropositive on ante-retroviral treatment?

Both conditions (quite prevalent in Uganda) are known to significantly affect glycation process, and with severe impact on values of HbA1c test in the diagnosis of diabetes mellitus.

2.3. Investigators reported in their methods section that continuous variables were to be reported in "mean +/- S.D."

However, in the results, most of the continuous variables were reported in "mean with 95% C.I." Even though there is a natural mechanism for conversion from one to another, consistency is key.

I would advice they choose one and leave the other in their reporting strategies.

2.4. Arbitrary Cut-off points: why was HbA1c level for abnormal glucose regulation (standard) made at 5.7% instead of the usual 5.6%?

- Can't that decision results to 'lower sensitivity' as stated in the manuscript?

- How different will it be if they stick to the standard HbA1c level cut-off point of 5.6%?

Notwithstanding, the manuscript is otherwise a good scientific report worth publishing after substantial corrections.

6. PLOS authors have the option to publish the peer review history of their article (what does this mean?). If published, this will include your full peer review and any attached files.

Reviewer #1: **Yes: **Kelvin Melkizedeck Leshabari

---

## [Author Response · Author response to Decision Letter 0]

25 Jun 2022

EDITORIAL COMMENTS

An interesting and relevant work to inform a key research gap.

The manuscript will benefit more by addressing the following:

Lin 205 to 208 authors will need to reference the table which describes these results. It is not clear as it is 155 participants are those with above 6mmol/L or "above or less."

Response: The 155 participants are those whose Fasting Plasma Glucose (FPG) results were greater than 6 mmol/L. We tested an equal number of participants whose results were less than or equal to 6 mmol/L. As advised, we have edited the manuscript to make this clearer and referenced Fig 1, which illustrates those results. These changes are on lines 239-243 of the manuscript’s marked-up version.

In the result section authors have described findings of the outcomes of the study before description of demographics and clinical characteristics. Authors will need to consistently follow the STARD guideline and use the STARD checklist to REWRITE their manuscript.

Response: We have corrected this error in the revised version of the protocol, moving the section at the beginning of the study outcomes from line 220 to 237. In this manuscript version, the results section begins with a description of the demographic and clinical characteristics. We have also rewritten the various sections of the manuscript according to STARD guidelines and completed STARD checklists for both the abstract and the body of the manuscript (see Supporting file 4 and Supporting file 5). The corresponding revisions made to the manuscript are as follows:

For the abstract, we have revised the subsections and content to reflect current recommendations in the “STARD for abstracts”. We have added a “Background and objectives” section (line 30 in the version of the manuscript with tracked changes) and renamed the “Conclusion” section “Discussion” (line 64). In the methods, we have identified the study as prospective (line 35), added the eligibility criteria (lines 36-37), noted that sampling was consecutive (line 36) and identified Glycated haemoglobin test (HBA1C) and FPG as index tests and Oral Glucose Tolerance Test (OGTT) as the clinical reference standard. In the results, we mention the number of participants with diabetes and those with Abnormal Glucose Regulation (AGR) included in the analysis (line 42). 

In the body of the manuscript, we have added the rationale for choosing OGTT as the clinical reference standard (lines 139-141). We have also explained that the index tests (FPG and HBA1C) were conducted before the OGTT in all cases (lines 162-163). In addition, we explained how we explored the sensitivity and specificity of the FPG and HBA1C would vary in a simultaneous testing scenario (lines 202-204). We moved the STARDS diagram to the results section (line 244) and updated it with the information on the number of participants with AGR according to the clinical reference standard.

We have also aligned the manuscript to the terminologies recommended by the STARD guidelines. Throughout the manuscript, we have replaced the word “gold standard” with “clinical reference standard” and referred to the HBA1C and FPG tests as “index tests”.

In table 2, authors are reporting comparable specificity (overlapping CI) however, in the result description and summary in the discussion, authors have reported low specificity of HBA1c as low specificity. Authors will need to clarify or rectify.

Indeed, the previous version of the manuscript had several inconsistencies regarding the interpretation of overlapping CIs. We have corrected this, noting that HBA1C sensitivity and specificity were comparable at both recommended and optimum cut-off points for diabetes. This is also true for AGR except for specificity at ADA recommended cut-off points, where it was lower for the HBA1C test [70.7% (95% CI 65.1-75.8)] and non-overlapping to that of the FPG test [93.5% (95% CI 88.6-96.4)]. These changes are on lines 43-58, 273, 280-283.

REVIEWER COMMENTS

Reviewer #1: The investigators reported in their manuscript that had applied 2-stage sampling strategies. However, in both cases, the approaches were non-probability sampling in nature. Strangely, the inferences were heavily dependent on probability sampling analysis. For all practical purposes, the so-called "consecutive sampling" is a non-probability sampling method. One of the immediate impacts (despite all posteriori statistical manoeuvres done) of such 'design error' include estimates that blow out disproportionately out of the bounded confidence interval values. That can be evident in their manuscript since optimum level specificity for their HbA1c had an estimate (98.7%) outside the stated 95% C.I. (96.2% - 98.4%) in both abstracts (refer to line 47 of page 2) and even in their table 3.

Usually, this thinking invalidates the entire analysis of their primary analysis.

However, I am optimistic about the findings, based on both 'near-infinitely large sample size', as well as 'inverse probability weighting', that controlled for bias in the primary findings (sensitivity, specificity, PPV & NPV), obtained out of likely 'partial verification bias' prominent by design. It is highly likely, that their large sample size approximate the 'real population' of interest and therefore, somehow offsets all the doubted possibilities of significant biases. A sensitivity analysis is likely warranted in this scenario. 

Response: Our decision is to use a consecutive was informed by the desire to enrol every eligible participant receiving care at the outpatient department. As such, our sample was highly representative of the accessible population [1]. In addition, the study was conducted for several months, ensuring that it is representative of the target population, i.e. patients accessible to ambulatory care at a district hospital. We, therefore, believe that our probability-based inferences and estimates are valid. 

Thank you for pointing out the out-of-range FPG specificity point estimate. Upon review, we realise that we used the NPV instead of the specificity to compute the corresponding 95% CIs. We have corrected this in the current version. The new 95% CI is 98.0-99.2. We have revised the current version of the manuscript (lines 54 and 276 of the tracked changes version) accordingly

We appreciate your observations regarding the large sample size as a strength for this study.

2. The study design was better suited as a population-based study instead of a hospital-based analysis.

2.1 - as stated by investigators, Iganga hospital is a referral health facility; hence, the studied sample is unlikely to represent the entire population.

There is a possibility that their findings are mere 'hospital statistics' data. Time after time, hospital statistics data are known to be 'negatively biased' with respect to the clinical diagnostic utility studies. No wonder, their sample has an age range of 30-75 years. Highly likely those 'older than 75 years cohort' who are the most at risk to be diabetic were selected out from participating in this study. Likewise, the 'younger cohort' (<30 years) who are otherwise, least likely to be impaired glucose tolerant/diabetic were unlikely to be part of this study

Response: Our decision to perform the study in an outpatient hospital population was a pragmatic choice motivated by our desire to inform opportunistic screening at the health facility level. This is probably a more realistic strategy than mass screening, which, in addition to being costly and out of reach for low and middle-income countries, may provide no added value regarding patients’ positive clinical outcomes. [2] It was never our intervention to make inferences about the general population. In our previous work on diabetes diagnosis, [3] we found that patients with diabetes often visit health facilities on multiple occasions in pursuit of a diagnosis but are not screened for diabetes. We, therefore, felt that evaluating test performance in an outpatient screening scenario would be a worthwhile investment to inform diabetes screening policy and practice in this setting. 

Limiting study participation to individuals no younger than 30 years was meant to minimise the potential recruitment of individuals with type 1 diabetes, which tends to develop earlier compared to type 2 diabetes, the focus of this study. For the upper limit of 75 years, we purposed to exclude the very elderly, as HBA1C may be inaccurate in this age category due to comorbidities that affect red blood cell lifespan. [4]

2.2. Prevalent clinical conditions in the population that can significantly affect HbA1c levels were not reported to have been controlled in the study by design.

For instance, how many (and proportion by %) of their participants in the study were pregnant women? and how many (and proportion by %) were HIV-seropositive on ante-retroviral treatment?

Both conditions (quite prevalent in Uganda) are known to affect the glycation process significantly and with a severe impact on values of the HbA1c test in the diagnosis of diabetes mellitus.

Response: We acknowledge the lack of data on the performance of the index tests, especially HBA1C, among HIV-positive individuals as a study limitation which we acknowledged in the original version of the manuscript (lines 370-378 in the current version with tracked changes). We have acknowledged the lack of data on the pregnancy status of the female participants in this study as another study limitation (lines 385-386). 

2.3. Investigators reported in their methods section that continuous variables were to be reported in "mean +/- S.D."

However, in the results, most of the continuous variables were reported in "mean with 95% C.I." Even though there is a natural mechanism for conversion from one to another, consistency is key.

I would advice they choose one and leave the other in their reporting strategies.

Response: Thanks for the observation regarding the inconsistency in reporting continuous variables. We have removed references to standard deviations (line 190 of the version with tracked changes).

2.4. Arbitrary Cut-off points: why was HbA1c level for abnormal glucose regulation (standard) made at 5.7% instead of the usual 5.6%?

- Can't that decision results to 'lower sensitivity' as stated in the manuscript?

- How different will it be if they stick to the standard HbA1c level cut-off point of 5.6%?

Response: According to the ADA, [5] the prediabetes range per the HBA1C test is 5.7-6.4%. This corresponds directly to the cut-off of ≥5.7%, which we used to compute and report HBA1C sensitivity and specificity in Table 3. According to this classification, a patient with an HBA1C score of 5.6% would be considered normoglycemic. Therefore, we believe that our results provide an accurate estimate of HBA1C performance in screening for AGR based on our data.

References

1. Roach KE. A clinician’s guide to specification and sampling. J Orthop Sports Phys Ther. 2001;31: 753–758. doi:10.2519/jospt.2001.31.12.753

2. Durao S, Ajumobi O, Kredo T, Naude C, Levitt NS, Steyn K, et al. Evidence insufficient to confirm the value of population screening for diabetes and hypertension in low- and-middle-income settings. S Afr Med J. 2015;105: 98–102. doi:10.7196/SAMJ.8819

3. Kasujja FX, Nuwaha F, Daivadanam M, Kiguli J, Etajak S, Mayega RW. Understanding the diagnostic delays and pathways for diabetes in eastern Uganda: A qualitative study. PLOS ONE. 2021;16: e0250421. doi:10.1371/journal.pone.0250421

4. LeRoith D, Halter JB. Diagnosis of Diabetes in Older Adults. Diabetes Care. 2020;43: 1373–1374. doi:10.2337/dci20-0013

5. ADA. Standards of Medical Care in Diabetes—2022 Abridged for Primary Care Providers. Clin Diabetes. 2022;40: 10–38. doi:10.2337/cd22-as01

---

## [Editor Report · Decision Letter 1]

21 Jul 2022

Glycated Haemoglobin and Fasting Plasma Glucose tests in the screening of outpatients for diabetes and abnormal glucose regulation in Uganda: a diagnostic accuracy study

PONE-D-22-06422R1

Dear Dr. Francis Xavier Kasujja,

We’re pleased to inform you that your manuscript has been judged scientifically suitable for publication and will be formally accepted for publication once it meets all outstanding technical requirements.

Kind regards,

Fredrick Baragi Haraka, MD, PhD

Academic Editor

PLOS ONE

Additional Editor Comments (optional):

Reviewers' comments:

None

---

## [Editor Report · Acceptance letter]

28 Jul 2022

PONE-D-22-06422R1 

Glycated Haemoglobin and Fasting Plasma Glucose tests in the screening of outpatients for diabetes and abnormal glucose regulation in Uganda: a diagnostic accuracy study 

Dear Dr. Kasujja:

I'm pleased to inform you that your manuscript has been deemed suitable for publication in PLOS ONE. Congratulations! Your manuscript is now with our production department. 

Kind regards, 

on behalf of

Dr. Fredrick Baragi Haraka 

Academic Editor

PLOS ONE